# The extent of algorithm aversion in decision-making situations with varying gravity

Ibrahim Filiz[1], Jan René Judek[1], Marco Lorenz[2]*, Markus Spiwoks[1]

**1** Faculty of Business, Ostfalia University of Applied Sciences, Wolfsburg, Germany, **2** Faculty of Economic Sciences, Georg August University Göttingen, Göttingen, Germany

* marco.lorenz@stud.uni-goettingen.de

**Data Availability Statement:** All relevant data are within the manuscript and its Supporting Information files.

**Funding:** The authors received no specific funding for this work.

## Abstract

Algorithms already carry out many tasks more reliably than human experts. Nevertheless, some subjects have an aversion towards algorithms. In some decision-making situations an error can have serious consequences, in others not. In the context of a framing experiment, we examine the connection between the consequences of a decision-making situation and the frequency of algorithm aversion. This shows that the more serious the consequences of a decision are, the more frequently algorithm aversion occurs. Particularly in the case of very important decisions, algorithm aversion thus leads to a reduction of the probability of success. This can be described as the tragedy of algorithm aversion.

## Introduction

Automated decision-making or decision aids, so-called algorithms, are becoming increasingly significant for many people's working and private lives. The progress of digitalization and the growing significance of artificial intelligence in particular mean that efficient algorithms have now already been available for decades (see, for example, [1]). These algorithms already carry out many tasks more reliably than human experts [2]. However, only a few algorithms are completely free of errors. Some areas of application of algorithms can have serious consequences in the case of a mistake–such as autonomous driving (cf. [3]), making medical diagnoses (cf. [4]), or support in criminal proceedings (cf. [5]). On the other hand, algorithms are also used for tasks which might not have serious consequences in the case of an error, such as dating service (cf. [6]), weather forecasts (cf. [7]), and the recommendation of cooking recipes (cf. [8]).

Some economic agents have a negative attitude towards algorithms. This is usually referred to as algorithm aversion (for an overview of algorithm aversion see [9,10]. Many decision-makers thus tend to delegate tasks to human experts or carry them out themselves. This is also frequently the case when it is clearly recognizable that using algorithms would lead to an increase in the quality of the results [11–13].

In decision-making situations which lead to consequences which are not so serious in the case of an error, a behavioral anomaly of this kind does not have particularly significant effects. In the case of a dating service, the worst that can happen is meeting with an unsuitable candidate. In the case of an erroneous weather forecast, unless it is one for seafarers, the worst that can happen is that unsuitable clothing is worn, and if the subject is the recommendation of

**Competing interests:** The authors have declared that no competing interests exist.

cooking recipes, the worst-case scenario is a bland meal. However, particularly in the case of decisions which can have serious consequences in the case of a mistake, diverging from the rational strategy would be highly risky. For example, a car crash or a wrong medical diagnosis can, in the worst case, result in someone's death. Being convicted in a criminal case can lead to many years of imprisonment. In these serious cases, it can be expected that people tend to think more thoroughly about what to do in order to make a reasonable decision. Can algorithm aversion be overcome in serious situations in order to make a decision which maximizes utility and which, at best, can save a life?

Tversky & Kahneman [14] show that decisions can be significantly influenced by the context of the decision-making situation. The story chosen to illustrate the problem influences the salience of the information, which can also lead to an irrational neglect of the underlying mathematical facts. This phenomenon is also referred to as the framing effect (for an overview see [15]). Irrespective of the actual probability of success, subjects do allow themselves to be influenced. This study therefore uses six mathematically identical decision situations with different contexts to examine whether the extent of algorithm aversion can be influenced by a framing effect.

Moreover, it is analyzed precisely which aspects of a decision affect the choice between algorithms and human experts the most. In particular, it is examined whether subjects are prepared to desist from their algorithm aversion in decision-making situations which can have severe consequences (three of the six scenarios). Expectancy theory [16] states that the importance of a task positively influences subjects' motivation in performing the task. Consistent with this, Mento, Cartledge & Locke [17] show in five experiments that increasing valence of a goal leads to higher goal acceptance and determination to achieve it. Gollwitzer [18] argues that the importance of a task determines the extent to which individuals develop and maintain commitment to the task. Similarly, Gendolla [19] asserts that "outcome valence and importance have effects on expectancy formation," where importance refers to the "personal relevance of events".

If algorithm aversion is due to decisions being made on gut instinct rather than analytically thought through, it should decrease with more meticulous expectancy formation, and increasing motivation and commitment, all of which result from task importance. We thus consider whether there are significantly different frequencies of algorithm aversion depending on whether the decision-making situations can have serious consequences or not.

## Literature review

Previous publications have defined the term algorithm aversion in quite different ways (Table 1). These different understandings of the term are reflected in the arguments put forward as well as in the design of the experiments carried out. From the perspective of some scholars, it is only possible to speak of algorithm aversion when an algorithm recognizably provides the option with the highest quality result or probability of success (cf. [10–12,20,21]). However, other scholars consider algorithm aversion to be present as soon as subjects exhibit a fundamental disapproval of an algorithm in spite of its possible superiority (cf. [22–28]).

Another important aspect of how the term algorithm aversion is understood is the question of whether and possibly also how the subjects learn about the superiority of an algorithm. Differing approaches were chosen in previous studies. Dietvorst, Simmons and Massey [12] focus on the gathering of experience in dealing with an algorithm in order to be able to assess its probability of success in comparison to one's own performance. In a later study, Dietvorst, Simmons and Massey [29] specify the average error of an algorithm. Alexander, Blinder and Zak [41] provide exact details on the probability of success of an algorithm, or they refer to the rate at which other subjects used an algorithm in the past.

**Table 1. Definitions of algorithm aversion in the literature.**

| Authors | Definition of algorithm aversion |
|---|---|
| Dietvorst, Simmons & Massey, 2015 [12] | "Research shows that evidence-based algorithms more accurately predict the future than do human forecasters. Yet when forecasters are deciding whether to use a human forecaster or a statistical algorithm, they often choose the human forecaster. This phenomenon, which we call *algorithm aversion (. . .)*" |
| Prahl & Van Swol, 2017 [28] | "The irrational discounting of automation advice has long been known and a source of the spirited "clinical versus actuarial" debate in clinical psychology research (Dawes, 1979; Meehl, 1954). Recently, this effect has been noted in forecasting research (Önkal et al., 2009) and has been called algorithm aversion (Dietvorst, Simmons, & Massey, 2015)." |
| Dietvorst, Simmons & Massey, 2018 [29] | "Although evidence-based algorithms consistently outperform human forecasters, people often fail to use them after learning that they are imperfect, a phenomenon known as *algorithm aversion*." |
| Commerford, Dennis, Joe & Wang, 2019 [30] | "(. . .) *algorithm aversion*–the tendency for individuals to discount computer-based advice more heavily than human advice, although the advice is identical otherwise." |
| Horne, Nevo, O'Donovan, Cho & Adali, 2019 [24] | "For example, Dietvorst et al. (Dietvorst, Simmons, and Massey 2015) studied when humans choose the human forecaster over a statistical algorithm. The authors found that aversion of the automated tool increased as humans saw the algorithm perform, even if that algorithm had been shown to perform significantly better than the human. Dietvorst et al. explained that aversion occurs due to a quicker decrease in confidence in algorithmic forecasters over human forecasters when seeing the same mistake occur (Dietvorst, Simmons, and Massey 2015)." |
| Ku, 2019 [21] | "(. . .) "algorithm aversion", a term refers by Dietvorst et al. (Dietvorst et al. 2015) means that humans distrust algorithm even though algorithm consistently outperform humans." |
| Leyer & Schneider, 2019 [31] | "In the particular context of the delegation of decisions to AI-enabled systems, recent findings have revealed a general algorithmic aversion, an irrational discounting of such systems as suitable decision-makers despite objective evidence (Dietvorst, Simmons and Massey, 2018)" |
| Logg, Minson & Moore, 2019 [25] | "(. . .) human distrust of algorithmic output, sometimes referred to as "algorithm aversion" (Dietvorst, Simmons, & Massey, 2015).[1] "; Footnote 1: "while this influential paper [of Dietvorst et al.] is about the effect that seeing an algorithm err has on people's likelihood of choosing it, it has been cited as being about how often people use algorithms in general." |
| Önkal, Gönül & De Baets, 2019 [32] | "(. . .) people are averse to using advice from algorithms and are unforgiving toward any errors made by the algorithm (Dietvorst et al., 2015; Prahl & Van Swol, 2017)." |
| Rühr, Streich, Berger & Hess, 2019 [26] | "Users have been shown to display an aversion to algorithmic decision systems [Dietvorst, Simmons, Massey, 2015] as well as to the perceived loss of control associated with excessive delegation of decision authority [Dietvorst, Simmons, Massey, 2018]." |
| Yeomans, Shah, Mullainathan & Kleinberg, 2019 [27] | "(. . .) people would rather receive recommendations from a human than from a recommender system (. . .). This echoes decades of research showing that people are averse to relying on algorithms, in which the primary driver of aversion is algorithmic errors (for a review, see Dietvorst, Simmons, & Massey, 2015)." |
| Berger, Adam, Rühr & Benlian, 2020 [33] | "Yet, previous research indicates that people often prefer human support to support by an IT system, even if the latter provides superior performance–a phenomenon called algorithm aversion." (. . .) "These differences result in two varying understandings of what algorithm aversion is: unwillingness to rely on an algorithm that a user has experienced to err versus general resistance to algorithmic judgment." |
| Burton, Stein & Jensen, 2020 [10] | "(. . .) algorithm aversion—the reluctance of human forecasters to use superior but imperfect algorithms—(. . .)" |

*(Continued)*

**Table 1.** (Continued)

| Authors | Definition of algorithm aversion |
|---|---|
| Castelo, Bos & Lehmann, 2020 [11] | "The rise of algorithms means that consumers are increasingly presented with a novel choice: should they rely more on humans or on algorithms? Research suggests that the default option in this choice is to rely on humans, even when doing so results in objectively worse outcomes." |
| De-Arteaga, Fogliato & Chouldechova, 2020 [34] | "*Algorithm aversion*–the tendency to ignore tool recommendations after seeing that they can be erroneous (. . .)" |
| Efendić, Van de Calseyde & Evans, 2020 [22] | "Algorithms consistently perform well on various prediction tasks, but people often mistrust their advice."; "However, repeated observations show that people profoundly mistrust algorithm-generated advice, especially after seeing the algorithm fail (Bigman & Gray, 2018; Diab, Pui, Yankelevich, & Highhouse, 2011; Dietvorst, Simmons, & Massey, 2015; Önkal, Goodwin, Thomson, Gönül, & Pollock, 2009)." |
| Erlei, Nekdem, Meub, Anand & Gadiraju, 2020 [35] | "Recently, the concept of algorithm aversion has raised a lot of interest (see (Burton, Stein, and Jensen 2020) for a review). In their seminal paper, (Dietvorst, Simmons, and Massey 2015) illustrate that human actors learn differently from observing mistakes by an algorithm in comparison to mistakes by humans. In particular, even participants who directly observed an algorithm outperform a human were less likely to use the model after observing its imperfections." |
| Germann & Merkle, 2020 [36] | "The tendency of humans to shy away from using algorithms even when algorithms observably outperform their human counterpart has been referred to as algorithm aversion." |
| Ireland, 2020 [37] | "(. . .) some researchers find that, when compared to humans, people are averse to algorithms after recording equivalent errors." |
| Jussupow, Benbasat & Heinzl, 2020 [38] | "(. . .) literature suggests that although algorithms are often superior in performance, users are reluctant to interact with algorithms instead of human agents–a phenomenon known as algorithm aversion" |
| Niszczota & Kaszás, 2020 [23] | "When given the possibility to choose between advice provided by a human or an algorithm, people show a preference for the former and thus exhibit algorithm aversion (Castelo et al., 2019; Dietvorst et al., 2015, 2016; Longoni et al., 2019)." |
| Wang, Harper & Zhu, 2020 [39] | "(. . .) people tend to trust humans more than algorithms even when the algorithm makes more accurate predictions." |
| Kawaguchi, 2021 [40] | "The phenomenon in which people often obey inferior human decisions, even if they understand that algorithmic decisions outperform them, is widely observed. This is known as algorithm aversion (Dietvorst et al. 2015)." |
| Köbis & Mossink, 2021 [20] | "When people are informed about algorithmic presence, extensive research reveals that people are generally averse towards algorithmic decision makers. This reluctance of "human decision makers to use superior but imperfect algorithms" (Burton, Stein, & Jensen, 2019; p.1) has been referred to as algorithm aversion (Dietvorst, Simmons, & Massey, 2015). In part driven by the belief that human errors are random, while algorithmic errors are systematic (Highhouse, 2008), people have shown resistance towards algorithms in various domains (see for a systematic literature review, Burton et al., 2019)." |

In addition, when dealing with algorithms, the way in which people receive feedback is of significance. Can subjects (by using their previous decisions) draw conclusions about the quality and/or success of an algorithm? Dietvorst, Simmons and Massey [12] merely use feedback in order to facilitate experience in dealing with an algorithm. Prahl and Van Swol [28] provide feedback after every individual decision, enabling an assessment of the success of the algorithm. Filiz et al. [42] follow this approach and use feedback after every single decision in order to examine the decrease in algorithm aversion over time.

Other aspects which emerge from the previous definitions of algorithm aversion in the literature are the reliability of an algorithm (perfect or imperfect), the observation of its reliability (the visible occurrence of errors), access to historical data on how the algorithmic forecast was drawn up; the setting (algorithm vs. expert; algorithm vs. amateur; algorithm vs. subject) as well as extent of the algorithm's intervention (does the algorithm act as an aid to decision-making or does it carry out tasks automatically?).

In our view, the superiority of an algorithm (higher probability of success) and the knowledge of this superiority are the decisive aspects. Algorithm aversion can only be present when subjects are clearly aware that not using an algorithm reduces the expected value of their utility and they do not deploy it nevertheless. A decision against the use of an algorithm which is known to be superior reduces the expected value of the subject's pecuniary utility and thus has to be viewed as a behavioral anomaly (cf. [43–45]).

## Methods and experimental design

We carry out an economic experiment in the laboratory of the Ostfalia University of Applied Sciences, in which the subjects assume the perspective of a businessperson who offers a service to his/her customers. A decision has to be made on whether this service should be carried out by specialized algorithms or by human experts.

The involvement of students as subjects was approved by the dean's office of the business faculty and the research commission of the Ostfalia University of Applied Sciences. The economic experiment took place as part of a regular laboratory class. All participants were at least 18 years of age at the time of the experiment and are therefore considered to be of legal age in Germany. The participants had confirmed their consent by registration for the economic experiment in the online portal of the Ostfalia University, which is sufficient according to the dean's office and the research commission. Before the start of the economic experiment, they were informed again that their participation was completely voluntary and that they could leave at any time.

In this framing approach, six decision-making scenarios are contrasted that entail different degrees of gravity of their potential consequences if they are executed not successfully. We base our experimental approach on the factual contexts in which algorithms can be used, described in the introduction, and assume that subjects perceive gravity differently in these contexts. The following services are considered: (1) Driving service with the aid of autonomous vehicles (algorithm) or with the aid of drivers, (2) The evaluation of MRI scans with the help of a specialized computer program (algorithm) or with the aid of doctors, (3) The evaluation of files on criminal cases with the aid of a specialized computer program (algorithm) or with the help of legal specialists, (4) A dating site providing matchmaking with the aid of a specialized computer program (algorithm) or with the support of staff trained in psychology, (5) The selection of recipes for cooking subscription boxes with the aid of a specialized computer program or the help of staff trained as professional chefs, and (6) The drawing up of weather forecasts with the help of a specialized computer program (algorithm) or using experienced meteorologists (Table 2).

The six scenarios that are part of this study were identified through a pre-test, in which additional scenarios were also presented from a literature analysis and brainstorming process. The final selection was made based on three criteria: comprehensibility (do the subjects understand what this application area for algorithms is about?), familiarity (do the subjects know the application area from personal experience or from the media?), and scope (are the high and low scope scenarios actually evaluated as such?). The scenarios are selected in such a way that they are relevant in the literature and that the subjects should be familiar with them from

**Table 2. Decision-making scenarios.**

| Decision-making scenarios |
| --- |
| (1) Driving service |
| (2) Evaluation of MRI scans |
| (3) The assessment of criminal case files |
| (4) Dating service |
| (5) Selection of cooking recipes |
| (6) Drawing up weather forecasts |

public debates or from their own experience. In this way, it is easier for the subjects to immerse themselves in the respective context. Detailed descriptions of the scenarios can be viewed in S3 File.

The study has a between-subjects design. Each subject is only confronted with one of a total of six scenarios. All six scenarios have the same probability of success: the algorithm carries out the service with a probability of success of 70%. The human expert carries out the service with a probability of success of 60%. The participants receive a show-up fee of €2, and an additional payment of €4 if the service is carried out successfully. Since we apply the same mathematical conditions of a successful execution of a service to each scenario, only the contextual framework of the six scenarios varies. A perfectly rational economic subject (homo oeconomicus) decides to use the algorithm in all six scenarios because this leads to the maximization of the expected value of the compensation. The context of the respective scenario does not play any role for a homo oeconomicus, because he exclusively strives to maximize his pecuniary benefit.

Before the experiment begins, all participants have to answer test questions (see S2 File). They have a maximum of two attempts at this. Participants who answer the test questions incorrectly twice are disregarded in the analysis, as the data should not be distorted by subjects who have misunderstood the task. The experiment starts with the participants being asked to assess the gravity of the shown decision-making scenario on a scale from 0 (not serious) to 10 (very serious). This allows us to evaluate the different scenarios based on the perceived gravity of the subjects. In this way, it is possible to assess how subjects perceive the potential effects in the context of one scenario compared to the context of another scenario. In the case of the driving service and the evaluation of MRI scans, it could be a matter of life and death. In the evaluation of documents in the context of criminal cases, it could lead to serious limitations of personal freedom. These scenarios could thus have serious consequences for third parties if they end unfavorably. The situation is different in the case of matchmaking, selecting cooking recipes and drawing up weather forecasts. Even when these tasks cannot be accomplished in a satisfactory way sometimes, the consequences should usually not be very serious. A date might turn out to be dull, or one is disappointed by the taste of a lunch, or you are out without a jacket in the rain. None of those things would be pleasant, but the potential consequences would be trivial.

A *homo oeconomicus* (a person who acts rationally in economic terms) must–regardless of the context–prefer the algorithm to human experts, because it maximizes his or her financial utility. Every decision in favor of the human experts has to be considered algorithm aversion.

Algorithm aversion is a phenomenon which can occur in a wide range of decision-making situations (cf. [10]). We thus presume that the phenomenon can also be observed in this study. Although the scenarios offer no rational reasons for choosing the human experts, some of the participants will do precisely this. Hypothesis 1 is: Not every subject will select the algorithm. Null hypothesis 1 is therefore: Every subject will select the algorithm.

There is some evidence that the extent of algorithm aversion is influenced by the framing of the conditions under which an algorithm operates. Hou & Jung [46] have subjects complete estimation tasks using algorithms. They vary the description of the algorithm using a framing approach and find that this has a significant impact on the willingness to follow the algorithm's advice. Utz, Wolfers & Göritz [47] investigate the perspective on a decision. In three scenarios, they use a framing approach to vary whether a subject is the decision maker or the one affected by the decision. The influence of perspective on the choice behavior between human and algorithm is significant only in one of the three scenarios, namely in the distribution of ventilators for Covid-19 patients.

Regarding the importance and consequences of a task, the findings to date are mixed. Castelo, Bos & Lehmann [11] use a vignette study to show that framing is suited to influencing algorithm aversion. A self-reported dislike for or distrust in algorithms appears to various degrees in different contexts of a decision. Likewise, Renier, Schmid Mast & Bekbergenova [48] study, among other things, the relationship between algorithm aversion and the magnitude of a decision for the human who must bear the consequences of the decision. In a vignette study, they vary the magnitude of the consequences that result from an algorithm error. According to their description of the task, the people affected may, for example, be wrongly denied a job contract or a loan. In contrast to Castelo, Bos & Lehmann [11], they conclude that the scope has no influence on the extent of algorithm aversion.

The difference in the results of the mentioned studies already shows that there still seems to be a large knowledge gap here. Sometimes a framing approach seems suitable to change decision behavior in the context of algorithm use, and sometimes not. Nonetheless, in all four studies mentioned above, the algorithm was not recognizably the most reliable alternative, and there is also no performance-related payment for the subjects. Algorithm aversion is therefore not modeled as a behavioral anomaly.

To extend our understanding, we analyze the extent of algorithm aversion in six differently framed decision situations. We believe that a clear financial incentive that models algorithm aversion as a behavioral anomaly will enhance the framing effect. We expect that the frame will have an influence on algorithm aversion analogous to Castelo, Bos & Lehmann [11] if the financial advantage of the algorithm is clearly recognizable. Hypothesis 2 is: The proportion of decisions made in favor of the algorithm will vary significantly between the decision situations perceived as serious and trivial. Null hypothesis 2 is therefore: The proportion of decisions made in favor of the algorithm will not vary significantly between the decision situations perceived as serious and trivial.

In the literature there are numerous indications that framing can significantly influence the decision-making behavior of subjects (cf. [14]). If subjects acted rationally and maximized their utility, neither algorithm aversion nor the framing effect would arise. Nonetheless, real human subjects–as the research in behavioral economics frequently shows–do not act like *homo oeconomicus*. Their behavior usually tends to correspond more to the model of bounded rationality put forward by Herbert A. Simon [49]. Human beings suffer from cognitive limitations–they fall back on rules of thumb and heuristics. But they do try to make meaningful decisions–as long as this does not involve too much effort. This kind of 'being sensible'–which is often praised as common sense–suggests that great efforts have to be made when decisions can have particularly serious consequences (for an overview of bounded rationality see [50–52]).

Jack W. Brehm's motivational intensity theory (see, e.g., [53]) identifies three main determinants of effort to make successful decisions: (1) The importance of the outcome of a successful decision, (2) the degree of difficulty of the task, and (3) the subjective assessment that the task can be successfully accomplished. The more important the outcome of a successful decision, the more pronounced the effort to make a successful decision. The more difficult the task

is in relation to the desired outcome and the lower the prospect of successfully accomplishing the task, the weaker the effort to make a successful decision is pronounced.

The last two aspects are unlikely to vary much across the six decision situations in this study. The degree of difficulty of the task is consistent in all six cases. All that is required is to weigh the algorithm's probability of success (70%) against the human expert's probability of success (60%). This is a simple task—in all six decision situations. It can be assumed that this level of difficulty is perceived as manageable by the subjects—in all six decision situations. However, the importance of the outcome of a decision differs in the six decision situations. Three decision situations have potentially serious consequences, and the other three decision situations have potentially trivial consequences. Thus, it is to be expected that subjects will try harder to make a successful decision in the decisions that involve potentially serious consequences. This is in line with other research that shows that the valence of a goal influences expectancy formation [19] and leads to increasing motivation [16] and commitment to a task [17,18].

This is also consistent with what would generally be recognized as common sense. This everyday common sense, which demands different levels of effort for decision-making situations with different degrees of gravity, could contribute towards the behavioral anomaly of algorithm aversion appearing more seldom in decisions with possible serious consequences than in decisions with relatively insignificant effects. The founding of a company is certainly given much more thought than choosing which television program to watch on a rainy Sunday afternoon. And much more care will usually be invested in the selection of a heart surgeon than in the choice of a pizza delivery service.

The assumption that higher valence of a situation leads to more effort in decision making has already been supported by experimental economics in other contexts. For example, Muraven & Slessareva [54] tell a subset of their subjects that their responses in an effort task will be used for important research projects to combat Alzheimer's disease. The mere belief that their effort may possibly reduce the suffering of Alzheimer's patients leads subjects to perform significantly better than in a control group. Since higher task importance may contribute to exerting more effort, we hypothesize that it also leads subjects to focus on the really relevant aspects of a decision (here: the different probabilities of success), thus eventually decreasing algorithm aversion. Hypothesis 3 is thus: The greater the gravity of a decision, the more seldom the behavioral anomaly of algorithm aversion arises. Null hypothesis 3 is therefore: Even when the gravity of a decision-making situation increases, there is no reduction in algorithm aversion.

## Results

This economic experiment is carried out between 2–14 November 2020 in the Ostfalia Laboratory of Experimental Economic Research (OLEW) of Ostfalia University of Applied Sciences in Wolfsburg. A total of 143 students of the Ostfalia University of Applied Sciences take part in the experiment. Of these, 91 subjects are male (63.6%), 50 subjects are female (35%) and 2 subjects (1.4%) describe themselves as non-binary. Of the 143 participants, 65 subjects (45.5%) study at the Faculty of Business, 60 subjects (42.0%) at the Faculty of Vehicle Technology, and 18 subjects (12.6%) at the Faculty of Health Care. Their average age is 23.5 years.

The experiment is programmed in z-Tree (cf. [55]). Only the lottery used to determine the level of success when providing the service is carried out by taking a card from a pack of cards. In this way we want to counteract any possible suspicion that the random event could be manipulated. The subjects see the playing cards and can be sure that when they choose the algorithm there is a probability of 70% that they will be successful (the pack of cards consists of seven +€4 cards and three ±€0 cards). In addition, they can be sure that if they choose a

human expert their probability of success is 60% (the pack of cards consists of six +€4 cards and four €±0 cards) (see S1 and S2 Figs).

The time needed for reading the instructions of the game (see S1 File), answering the test questions (see S2 File), and carrying out the task is 10 minutes on average. A show-up fee of €2 and the possibility of a performance-related payment of €4 seem appropriate for the time spent—it is intended to be sufficient incentive for meaningful economic decisions, and the subjects do give the impression of being concentrated and motivated.

We provide the subjects with only one contextual framework of a decision situation at a time. Here, the subjects are presented with decision situations in the context of driving service (25 subjects), evaluating MRI scans (24 subjects), assessing criminal case files (22 subjects), dating service (24 subjects), selecting cooking recipes (23 subjects), and drawing of weather forecasts (25 subjects). Subjects were distributed similarly evenly across the contextual decision situations in terms of gender and faculty.

Overall, only 87 out of 143 subjects (60.84%) decide to delegate the service to the (superior) algorithm. A total of 56 subjects (39.16%) prefer to rely on human experts in spite of the lower probability of success. Null hypothesis 1 thus has to be rejected. The result of the chi-square goodness of fit test is highly significant ($\chi^2$ (N = 143) = 21.93, p < 0.001). On average, around two out of five subjects thus tend towards algorithm aversion. All subjects should be aware that preferring human experts and rejecting the algorithm reduces the expected value of the performance-related payment. However, the need to decide against the algorithm is obviously strong in a part of the subjects. To investigate the effects of our framing approach on algorithm aversion (hypothesis 2), we must first examine how subjects evaluate the six differently framed scenarios. The subjects perceive the gravity of the decision situations differently (Fig 1). While in the contextual decision situations driving service (mean gravity of 8.88), evaluation of MRI scans (mean gravity of 9.42) and assessment of criminal case files (mean gravity of 8.68), the potential consequences of not successfully performing the service are perceived as comparatively serious, the contextual decision situations dating service (mean gravity of 6.33), selection of cooking recipes (mean gravity of 7.87) and drawing up weather forecasts (mean gravity of 5.52) show less pronounced perceived gravity of potential consequences (Table 3).

In the six scenarios, each with identical chances of success for execution by a human expert or an algorithm, subjects decide by whom the service should be performed depending on the context in which the situation is presented. By considering the context, subjects arrive at an assessment of the gravity of the potential consequences of not successfully performing the service (Fig 1). Even though the six scenarios differ considerably from each other in their context, they are also similar in the assessment of their gravity when viewed individually. The perceived gravity of the scenarios as reported by the subjects suggests that the decision situations can be considered in two clusters, decisions with possibly serious consequences (for the highest mean gravity scores) and decisions with possibly trivial consequences (for the lowest mean gravity scores).

The comparison of the contextual decision situations with possibly serious consequences and those with possibly trivial consequences, as indicated by the mean values of the gravity levels per decision situation, show that the perceived gravity of the six scenarios is (highly) significantly different when using the Wilcoxon rank-sum test (Table 4). For example, perceived gravity of driving service differs from dating service with p < 0.001. Only the scenarios driving service and assessment of criminal case files differ from the scenario recipe selection only at a significance level of 0.1, as already suggested by their mean gravity. On average, the possible consequences of recipe selection are perceived as slightly more serious, but not as serious as driving service, evaluating MRI scans or criminal case files. Cohen's d shows how much the means of two samples differ. An effect size of 0.2 corresponds to small effects, 0.5 to medium effects, and 0.8 to large effects [56].

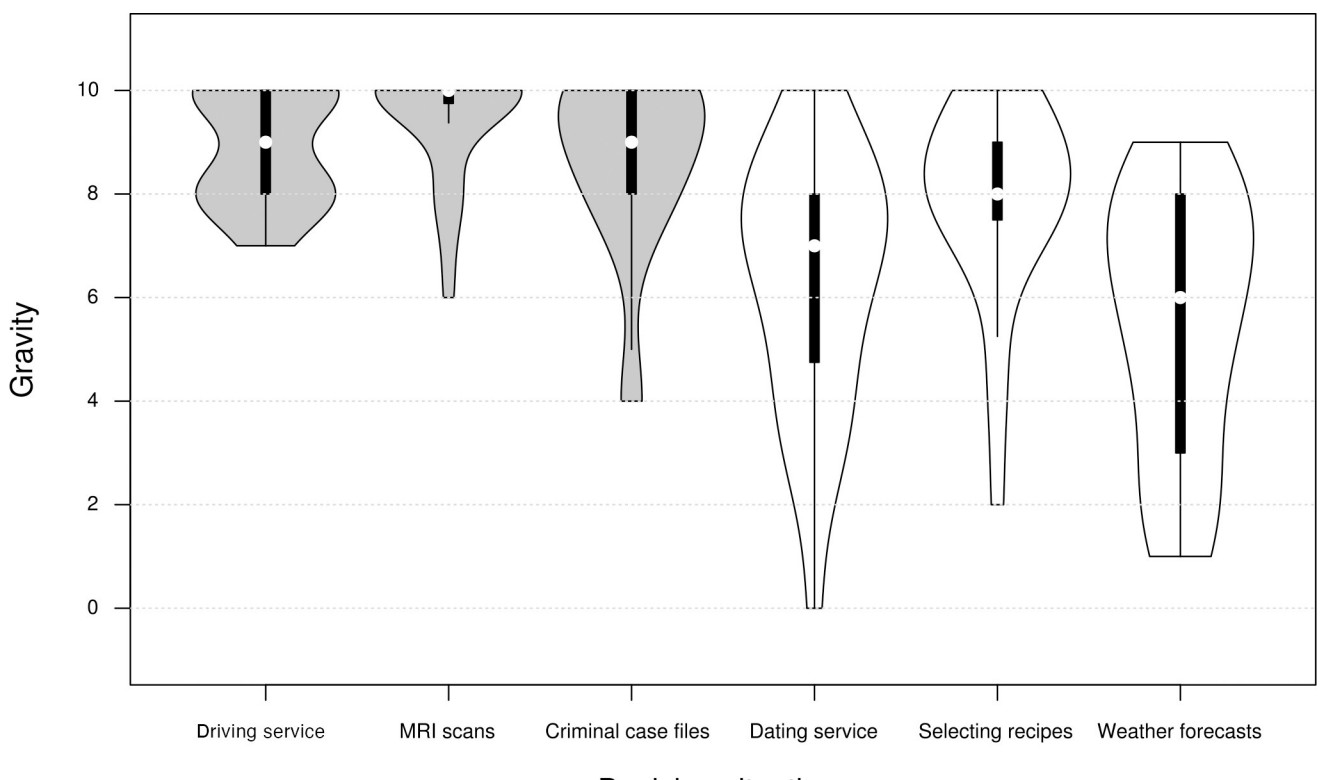

**Fig 1. Violin plots for the assessment of the gravity of the decision-making situations.**

This confirms that subjects perceive the consequences of decision contexts to vary in gravity and leads to the classification of decision contexts into cluster A1 (possibly serious consequences: driving service, evaluation of MRI scans, and criminal case files) and cluster A2 (possibly trivial consequences: dating service, selecting cooking recipes, and weather forecasts) that we propose in this framework. The violin plot of the summarized decision situations shows that the subjects rate the gravity higher in contexts with critical consequences for physical integrity than in contexts where it does not matter (Fig 2). However, a direct comparison of the violin plots also shows that the range of decision situations rated as having trivial consequences is wider than that of the others, since some subjects also rate the gravity as very high here.

Still, the possible consequences of each decision situation from cluster A1 are rated by the subjects as more serious than those from cluster A2. According to this classification, the mean

**Table 3. Evaluation of gravity in a contextual decision situation.**

| Scenario | # | Mean value of gravity | Median | Standard deviation |
|---|---|---|---|---|
| (1) Driving service | 25 | 8.88 | 9 | 1.09 |
| (2) Evaluation of MRI scans | 24 | 9.42 | 10 | 1.14 |
| (3) Criminal case files | 22 | 8.68 | 9 | 1.78 |
| (4) Dating service | 24 | 6.33 | 7 | 2.50 |
| (5) Selection of recipes | 23 | 7.87 | 8 | 1.96 |
| (6) Weather forecasts | 25 | 5.52 | 6 | 2.57 |

**Table 4. Comparison of perceived gravity using Wilcoxon rank-sum test and Cohen's d.**

| | (4) Dating service | | (5) Selection of recipes | | (6) Weather forecasts | |
|---|---|---|---|---|---|---|
| | p-value[*] | Cohen's d | p-value[*] | Cohen's d | p-value[*] | Cohen's d |
| (1) Driving service | 0.000 | 1.33 | 0.064 | 0.64 | 0.000 | 1.70 |
| (2) Evaluation of MRI scans | 0.000 | 1.59 | 0.000 | 0.97 | 0.000 | 1.95 |
| (3) Criminal case files | 0.000 | 1.07 | 0.061 | 0.43 | 0.000 | 1.41 |

[*]p-values using Wilcoxon rank-sum test.

of the perceived gravity for the decision situations with possibly serious consequences (A1) is 9.0 with a standard deviation of 1.37. In contrast, when the gravity of the decision situations with possibly trivial consequences (A2) is evaluated, the mean is 6.54 with a standard deviation of 2.53 (Table 5). The Wilcoxon rank-sum test shows that the gravity of the decision situations in cluster A1 is assessed as significantly higher than that of the decision situations in cluster A2 ($z = 6.689$; $p < 0.001$).

Furthermore, a difference in the number of decisions in favor of the algorithm between the two clusters can be observed. While 50.7% of the subjects in cluster A1 choose the algorithm, 70.83% in cluster A2 rely on it (for the individual decisions in the contextual decision situations, see Table 6). The chi-square test reveals that null hypothesis 2 has to be rejected ($\chi^2$ (N = 143) = 6.08, p = 0.014). The frequency with which algorithm aversion occurs is influenced by the implications involved in the decision-making situation. The framing effect has an impact.

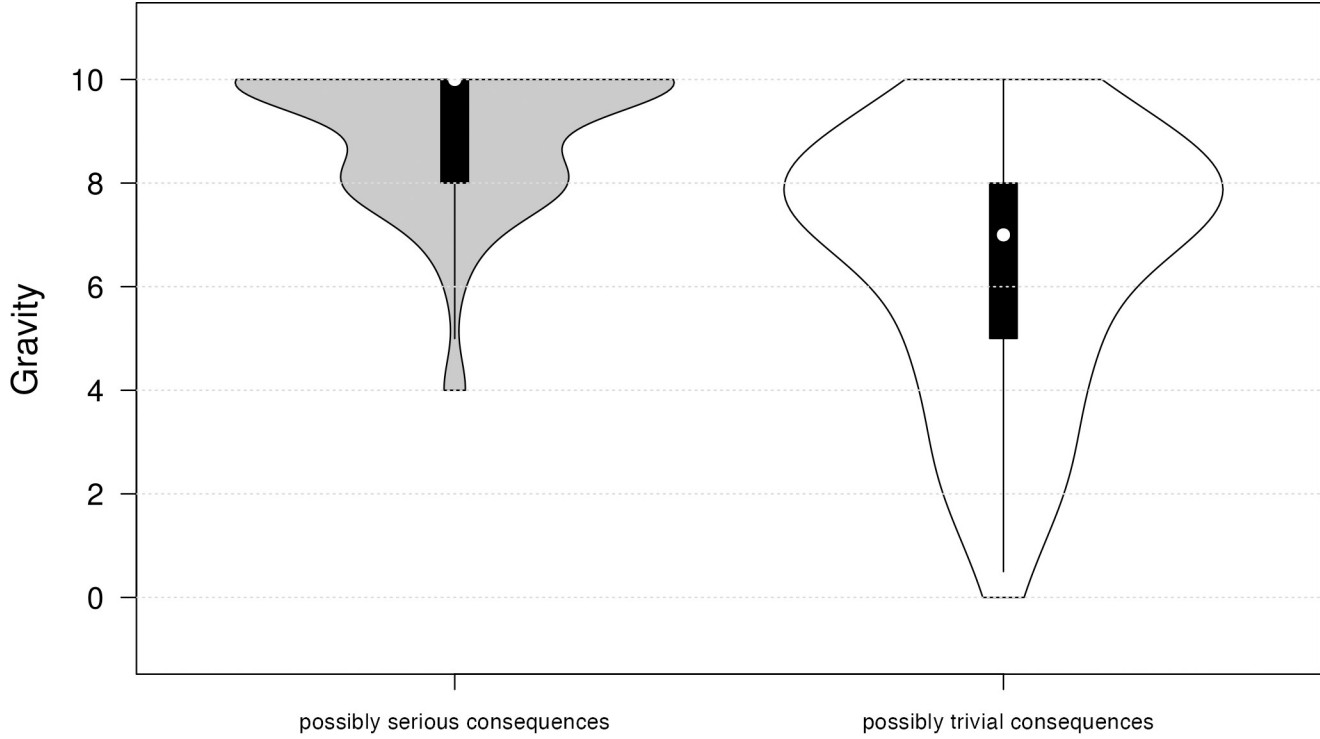

**Fig 2. Violin plots for scenarios with possibly serious and trivial consequences.**

**Table 5. Evaluation of gravity in clusters A1 and A2.**

|  | Cluster A1 (serious) | Cluster A2 (trivial) |
|---|---|---|
| Median | 10 | 7 |
| Average0} | 9.00 | 6.54 |
| Standard deviation | 1.37 | 2.53 |

There may be situations in which people like to act irrationally at times. However, common sense suggests that one should allow oneself such lapses in situations where serious consequences must not be feared. For example, there is a nice barbecue going on and the host opens a third barrel of beer although he suspects that this will lead to hangovers the next day among some of his guests. In the case of important decisions, however, one should be wide awake and try to distance oneself from reckless tendencies. For example, if the same man visits a friend in hospital whose life would be acutely threatened by drinking alcohol after undergoing a complicated stomach operation, he would be wise to avoid bringing him a bottle of his favorite whisky. This comparison of two examples illustrates what could be described as common sense and would be approved of by most neutral observers.

Nevertheless, the results of the experiment point in the opposite direction. A framing effect sets in, but not in the way one might expect. Whereas in cluster A1 (possibly serious consequences) 49.3% of the subjects do exhibit the behavioral anomaly of algorithm aversion, this is only the case in 29.17% of the subjects in cluster A2 (possibly trivial consequences) (Table 6). It seems that algorithm aversion is all the more pronounced in important tasks.

This is confirmed by a regression analysis which demonstrates the relationship between algorithm aversion and the perceived gravity of a scenario. To perform the regression analysis, we detach from the pairwise consideration of the two clusters and relate how serious an economic agent perceived the potential consequences of his or her decision and how it was decided. This is independent of the decision context, only the perceived gravity and the associated decision are considered. For the possible assessments of the consequences (from 0 = not serious to 10 = very serious), the respective average percentage of the decisions in favor of the algorithm is determined. The decisions of all 143 subjects are included in the regression analysis (Fig 3).

If the common sense described above would have an effect, the percentage of decisions for the algorithm from left to right (in other words with increasing perceived gravity of the decision-making situation) would tend to rise. Instead, the opposite can be observed. Whereas in

**Table 6. Decisions for and against the algorithm.**

|  |  | Total | Decisions for the algorithm | | Decisions against the algorithm | |
|---|---|---|---|---|---|---|
|  |  |  | n | Percent | n | Percent |
| **Cluster A1** | **(serious)** | **71** | **36** | **50.70%** | **35** | **49.30%** |
| (1) Driving car |  | 25 | 13 | 52.00% | 12 | 48.00% |
| (2) Evaluation of MRI scans |  | 24 | 13 | 54.17% | 11 | 45.83% |
| (3) Criminal case files |  | 22 | 10 | 45.45% | 12 | 54.55% |
| **Cluster A2** | **(trivial)** | **72** | **51** | **70.83%** | **21** | **29.17%** |
| (4) Dating service |  | 24 | 18 | 75.00% | 6 | 25.00% |
| (5) Selection of recipes |  | 23 | 14 | 60.87% | 9 | 39.13% |
| (6) Weather forecasts |  | 25 | 19 | 76.00% | 6 | 24.00% |
| Σ | **(total)** | **143** | **87** | **60.84%** | **56** | **39.16%** |

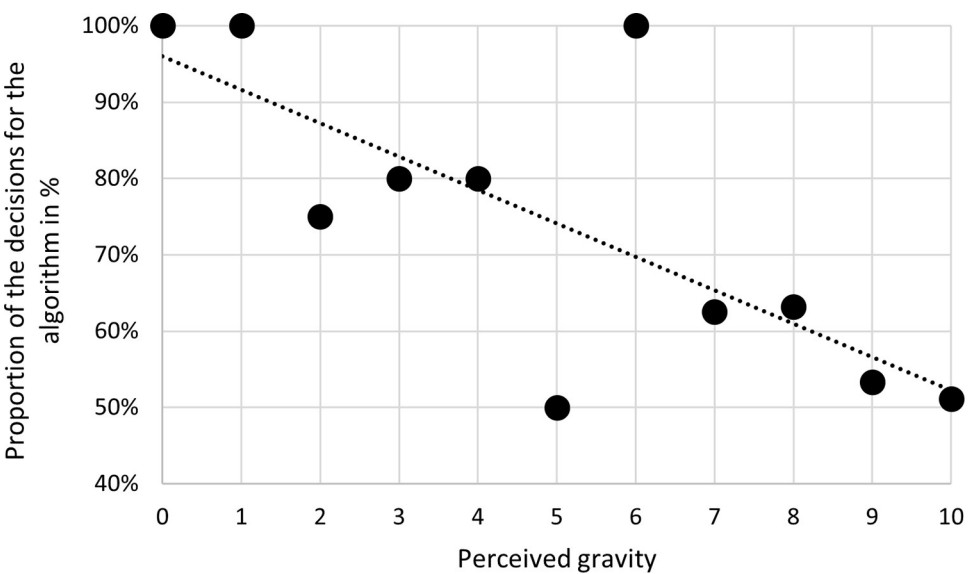

**Fig 3. Decisions in favor of the algorithm depending on the perceived gravity of the decision-making situation.**

the case of only a low level of gravity (zero and one) 100% of decisions are still made in favor of the algorithm, the proportion of decisions for the algorithm decreases with increasing gravity. In the case of very serious implications (nine and ten), only somewhat more than half of the subjects decide to have the service carried out by an algorithm (Fig 3). If the perceived gravity of a decision increases by a unit, the probability of a decision in favor of the algorithm falls by 3.9% (t = -2.29; p = 0.023). The 95% confidence interval ranges from -7.27% to -0.54%. Null hypothesis 3 can therefore not be rejected. In situations which might have serious consequences in the case of an error, algorithm aversion is actually especially pronounced.

Further analysis shows that the choices between algorithms and human experts are also not statistically significantly influenced by gender ($\chi^2$ (N = 143) = 2.22, p = 0.136), age (t (N = 143) = -0.44, = 0.661), mother tongue ($\chi^2$ (N = 143) = 2.68, p = 0.102), faculty at which a subject is studying ($\chi^2$ (N = 143) = 1.06, p = 0.589), semester (t (N = 143) = 0.63, p = 0.528), or previous participations in economic experiments ($\chi^2$ (N = 143) = 0.21, p = 0.644).

The six scenarios differ in numerous aspects. In order to identify the main factors influencing the decisions of the subjects, clusters are formed based on different criteria and examined with regard to differences in the subjects' selection behavior. There are a total of ten ways to divide six scenarios into two clusters. All ten clustering opportunities are shown in Table 7.

The criterion in focus in this study is the scope of a decision (clustering opportunity). We can group the six scenarios into tasks that have potentially serious consequences if performed incorrectly, e.g., death or unjust imprisonment. These are mainly driving service, evaluation of MRI scans, and assessment of criminal case files (cluster A1). On the other hand, there are tasks where the consequences are trivial if performed poorly. These are dating service, selection of cooking recipes, and weather forecasts (cluster A2). The chi-square test shows that the willingness to use an algorithm is significantly higher in the latter cluster ($\chi^2$ (N = 143) = 6.08, p = 0.014). The more serious the consequences of a decision, the less likely subjects are to delegate the decision to an algorithm.

Another aspect is the familiarity with a task (clusters J). A connection between algorithm aversion and familiarity has been suspected for some time. Luo et al. [57] argue that the more familiar and confident sales agents in dealing with a task, the more pronounced their

**Table 7. Overview of all possible clusters obtained by grouping the frames evenly.**

| Clustering Opportunities | Cluster | Frames | n | Algorithm Use | $\chi^2$ | p-value |
|---|---|---|---|---|---|---|
| A | A1<br>A2 | (1) (2) (3)<br>(4) (5) (6) | 71<br>72 | 50.70%<br>70.83% | 6.080 | **0.014** |
| B | B1<br>B2 | (1) (2) (4)<br>(3) (5) (6) | 73<br>70 | 60.27%<br>61.43% | 0.020 | 0.888 |
| C | C1<br>C2 | (1) (2) (5)<br>(3) (4) (6) | 72<br>71 | 55.56%<br>66.20% | 1.699 | 0.192 |
| D | D1<br>D2 | (1) (2) (6)<br>(3) (4) (5) | 74<br>69 | 60.81%<br>60.87% | 0.000 | 0.994 |
| E | E1<br>E2 | (1) (3) (4)<br>(2) (5) (6) | 71<br>72 | 57.75%<br>63.89% | 0.566 | 0.452 |
| F | F1<br>F2 | (1) (3) (5)<br>(2) (4) (6) | 70<br>73 | 52.86%<br>68.49% | 3.667 | 0.056 |
| G | G1<br>G2 | (1) (3) (6)<br>(2) (4) (5) | 72<br>71 | 58.33%<br>63.38% | 0.382 | 0.536 |
| H | H1<br>H2 | (1) (4) (5)<br>(2) (3) (6) | 72<br>71 | 62.50%<br>59.16% | 0.168 | 0.682 |
| I | I1<br>I2 | (1) (4) (6)<br>(2) (3) (5) | 74<br>69 | 67.57%<br>53.62% | 2.914 | 0.088 |
| J | J1<br>J2 | (1) (5) (6)<br>(2) (3) (4) | 73<br>70 | 63.01%<br>58.57% | 0.296 | 0.586 |

(1) = Driving service, (2) = Evaluation of MRI scans, (3) = Assessment of criminal case files, (4) = Dating service, (5) = Selection of cooking recipes, (6) = Weather forecasts.

algorithm aversion is. Gaube et al. [58] explicitly examine the influence of familiarity with a task on physicians' algorithm aversion in the context of evaluating human chest radiographs. They contrast experienced radiologists, who have a great deal of routine with this task, with inexperienced emergency physicians. Their results also suggest that algorithm aversion may increase with increasing experience in handling a task. We can group the six scenarios into tasks that are performed frequently, perhaps even daily, by an average person. These are driving a car, selection of cooking recipes, and weather forecasts. Almost every day, each of us commutes to work or other places, decides what to eat, and wonders what the weather will be like during the day. On the other hand, evaluation of MRI scans and assessment of criminal case files are activities that most of us may never have encountered, and dating service is something that those who are single may use from time to time, and those who are in a relationship (hopefully) not that much. The chi-square test shows no significant difference between the clusters J1 and J2 ($\chi^2$ (N = 143) = 0.30, p = 0.586). Thus, the willingness to use an algorithm does not seem to be considerably affected by how often we engage in a particular activity.

Further interesting aspects are whether an algorithm requires an expert to operate it adequately or whether it can also be used by a layperson (cluster H), whether a task requires human skills, such as empathy, or not (Cluster D), and the maturity of the technology, i.e., whether the use of algorithms is already widespread today or not (Cluster F). Algorithm aversion has been observed both in extremely simple algorithms that a layperson can easily operate by him- or herself and in extremely complex algorithms (numerous examples can be found in [11]). Regarding human skills, Fuchs et al. [59] find that algorithm aversion is particularly high for tasks that are driven more by human skills than by mathematical data analysis. Kaufmann [60] shows that algorithm aversion can occur to a large extent in student performance evaluation, a task that is characterized as requiring a lot of empathy. On the maturity of technology,

already 17 years ago Nadler & Shestowsky [61] raise the question of whether subjects may become accustomed to using an algorithm the longer it is established in the market.

It turns out that the willingness to choose an algorithm does not depend on the amount of expertise required to operate it ($\chi^2$ (N = 143) = 0.17, p = 0.682), nor on the extent to which human skills are involved in the task it is supposed to perform ($\chi^2$ (N = 143) = 0.00, p = 0.994). Regarding the maturity of technology, we see that activities that are already automated frequently in practice today, such as making weather forecasts, are also delegated to the algorithm much more often in the experiment ($\chi^2$ (N = 143) = 3.67, p = 0.056). However, the difference between the clusters F1 and F2 is not as large as between the clusters A1 and A2. Moreover, there is also no significant difference at a significance level of 0.05 in the frequency with which an algorithm is selected in the remaining five clustering opportunities. It therefore seems that of all the differences between the frames, the gravity of consequences of a decision are the most important aspect.

## Discussion

### General

The results are surprising, given that common sense would deem–particularly in the case of decisions which might have serious consequences–that the option with the greatest probability of success should be chosen. Also, with regard to Brehm's motivational intensity theory, it can be argued that the importance of the successful execution of the action is not adequately reflected in the subjects' decisions. In line with Hou & Jung [46] and Castelo, Bos & Lehmann [11], our results also show that a framing approach is suitable to influence decisions to engage an algorithm. The study by Utz, Wolfers & Göritz [47] shows that the preference to use an algorithm in moral scenarios (distribution of ventilators for Covid-19 treatment) is low. In our study, in scenarios that were perceived as scenarios with potentially serious consequences (driving service, evaluation of MRI scans and criminal case files) and also raise moral issues, a lower utilization rate of the algorithm is also shown. A survey by Grzymek & Puntschuh [62] clearly shows that people are less likely to use an algorithm in decision-making situations with potentially serious consequences, such as diagnosing diseases, evaluating creditworthiness, trading stocks, or pre-selecting job applicants, but more likely to use an algorithm in scenarios with less serious consequences, such as spell-checking, personalizing advertisements, or selecting the best travel route. In contrast, Renier, Schmid Mast & Bekbergenova [48] found no effect of gravity on the extent to which participants demand an improvement to an algorithm.

If subjects allow themselves to be influenced by algorithm aversion to make decisions to their own detriment, they should only do so when they can take responsibility for the consequences with a clear conscience. In cases where the consequences are particularly serious, maximization of the success rate should take priority. But the exact opposite is the case. Algorithm aversion appears most frequently in cases where it can cause the most damage. To this extent it seems necessary to speak of the tragedy of algorithm aversion.

### Implications

Our results suggest that algorithm aversion is particularly prevalent where potential errors have dire consequences. This means that algorithm aversion should be especially addressed by those developers, salespeople, and other staff whose supervised areas of operation are related to human health and safety. This can be done, for example, through staff training and intensive field testing with potential users. In addition, the results suggest that clever framing of the activity that an algorithm undertakes can make users more likely to use the algorithm. For

example, neutral words should be chosen when advertising medical or investment algorithms, rather than unnecessarily pointing out the general risks of such activity.

### Limitations and directions for future research

Despite their advantages in establishing causal relationships, framing studies always carry the risk that subjects may have many other associations with the decision-making situations that are not the focus of the study, and yet have an unintended influence on the results. In our study, these are in particular the complexity and subjectivity of the tasks, but also moral aspects, that may be more relevant in the decisions with potentially serious consequences, in which the physical well-being or the freedom of humans are at stake. In addition, we do not focus on the variation in perceived gravity within one scenario (e.g., MRI scans for live threatening diagnosis vs. MRI scans for less severe diagnosis), but rather on the variation in gravity between different scenarios, which could be a risk in regards to causality. It remains for further studies to vary the gravity within one scenario. Moreover, these aspects may also include the familiarity from the everyday experiences of the subjects, which should be higher, for example, for weather forecasts than for MRI scans. However, these associations do not affect our core result. Our regression analysis only considers the correlation between algorithm aversion and the subjectively perceived gravity of consequences, regardless of the scenario, and finds that higher perceived consequences in general lead to an increase in algorithm aversion.

Second, it should be noted that prior experiences of the subjects and the maturity of the technologies may lead to different expectations regarding the success rates. For example, the use of algorithms for weather forecasting is already advanced and it is to be expected that an algorithm would perform better here than a human. In autonomous driving, on the other hand, the technology is not yet as advanced. However, to ensure the comparability of the scenarios, in our framing approach the probabilities must be identical in all scenarios, which may not always fit the subjects' expectations. In addition, the success rates are directly given in the instructions of our experiment. In real life, we would first gather our own experience in all these areas to get an idea of when to rely on algorithms and when not to. Moreover, the sample size of our experiment is rather small with 143 participants. We therefore encourage future research efforts to further explore our results in a research design with more practice-oriented conditions and with a larger sample.

Finally, it is needless to say that the consequences of the decisions in our experiments might have to be borne by third parties. It would be possible to continue this line of research by giving up the framing approach and modeling a situation where the subjects are directly affected. In this case, different incentives would have to be introduced into the decision situations. Success in scenarios with possible serious consequences would then have to be rewarded with a higher amount than in scenarios with trivial consequences. However, we presume that our results would also be confirmed by an experiment based on this approach, given that it is a between-subjects design in which every subject is only presented with one scenario. Whether one receives €4 or €8 for a successful choice will probably not have a notable influence on the results. Nonetheless, the empirical examination of this assessment is something which will have to wait for future research efforts.

### Summary

Many people decide against the use of an algorithm even when it is clear that the algorithm promises a higher probability of success than a human mind. This behavioral anomaly is referred to as algorithm aversion.

The subjects are placed in the position of a businessperson who has to choose whether to have a service carried out by an algorithm or by a human expert. If the service is carried out successfully, the subject receives a performance-related payment. The subjects are informed that using the respective algorithm leads to success in 70% of all cases, while the human expert is only successful in 60% of all cases. In view of the recognizably higher success rate, there is every reason to trust in the algorithm. Nevertheless, just under 40% of the subjects decide to use the human expert and not the algorithm. In this way they reduce the expected value of their performance-related payment and thus manifest the behavioral anomaly of algorithm aversion.

The most important objective of the study is to find out whether decision-making situations of varying gravity can lead to differing frequencies of the occurrence of algorithm aversion. To do this, we choose a framing approach. Six decision-making situations (with potentially serious / trivial consequences) have an identical payment structure. Against this background there is no incentive or reason to act differently in each of the six scenarios. It is a between-subjects approach–each subject is only presented with one of the six decision-making situations.

In the three scenarios with potentially serious consequences for third parties, just under 50% of the subjects exhibit algorithm aversion. In the three scenarios with potentially trivial consequences for third parties, however, less than 30% of the subjects exhibit algorithm aversion.

This is a surprising result. If a framing effect were to occur, it would have been expected to be in the opposite direction. In cases with implications for freedom or even danger to life, one should tend to select the algorithm as the option with a better success rate. Instead, algorithm aversion shows itself particularly strongly here. If it is only a matter of arranging a date, creating a weather forecast or offering cooking recipes, the possible consequences are quite clear. In a situation of this kind, one can still afford to have irrational reservations about an algorithm. Surprisingly, however, algorithm aversion occurs relatively infrequently in these situations.

One can call it the tragedy of algorithm aversion because it arises above all in situations in which it can cause particularly serious damage.

## Supporting information

**S1 Fig.**
(TIF)

**S2 Fig.**
(TIF)

**S1 Data. Framing and algorithm aversion—Supplementary data.**
(XLSX)

**S1 File. Instructions for the game.**
(DOCX)

**S2 File. Test questions.**
(DOCX)

**S3 File. Decision-making situations.**
(DOCX)

**S4 File. Determination of the random event with the aid of a lottery.**
(DOCX)

## Acknowledgments

The authors would like to thank the editor, the anonymous reviewers, the participants in the German Association for Experimental Economic Research e.V. (GfeW) Meeting 2021, the participants in the Economic Science Association (ESA) Meeting 2021, and the participants in the PhD seminar at the Georg August University of Göttingen, for their constructive comments and useful suggestions, which were very helpful in improving the manuscript.

## Author Contributions

**Conceptualization:** Ibrahim Filiz, Jan René Judek, Marco Lorenz, Markus Spiwoks.

**Data curation:** Ibrahim Filiz, Jan René Judek, Marco Lorenz, Markus Spiwoks.

**Formal analysis:** Ibrahim Filiz, Jan René Judek, Marco Lorenz, Markus Spiwoks.

**Investigation:** Ibrahim Filiz, Jan René Judek, Marco Lorenz, Markus Spiwoks.

**Methodology:** Ibrahim Filiz, Jan René Judek, Marco Lorenz, Markus Spiwoks.

**Project administration:** Ibrahim Filiz, Jan René Judek, Marco Lorenz, Markus Spiwoks.

**Resources:** Ibrahim Filiz, Jan René Judek, Marco Lorenz, Markus Spiwoks.

**Software:** Ibrahim Filiz, Jan René Judek, Marco Lorenz, Markus Spiwoks.

**Supervision:** Ibrahim Filiz, Jan René Judek, Marco Lorenz, Markus Spiwoks.

**Validation:** Ibrahim Filiz, Jan René Judek, Marco Lorenz, Markus Spiwoks.

**Visualization:** Ibrahim Filiz, Jan René Judek, Marco Lorenz, Markus Spiwoks.

**Writing – original draft:** Ibrahim Filiz, Jan René Judek, Marco Lorenz, Markus Spiwoks.

**Writing – review & editing:** Ibrahim Filiz, Jan René Judek, Marco Lorenz, Markus Spiwoks.

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
