## [Decision Letter · Decision Letter 0]

10 Mar 2022

PONE-D-21-39459The Tragedy of Algorithm AversionPLOS ONE

Dear Dr. Lorenz,

Thank you for submitting your manuscript to PLOS ONE. After careful consideration, we feel that it has merit but does not fully meet PLOS ONE’s publication criteria as it currently stands. Therefore, we invite you to submit a revised version of the manuscript that addresses the points raised during the review process.

 Both reviewers agree that the question you are tackling is important. However, they adress some important issues with the paper, too.From my own reading and the comments of the reviewers, I would suggest you look carefully at the recommendations provided that focus on the link with theory. This is something that can be improved rather easily, but should be done in a clear  and sensible way.The issues relating the design are more difficult to address. I agree with reviewer 2 that there are more differences between the conditions rather than just gravity of consequences. As your sample size is also not convincing, I am somewhat inclined to suggest you run more experiments, making it a series of experiments by that. You could keep your first set of experiments, but add a new set that tries to handle the issues reviewer 2 is rising. As reviewer 2 also suggests that you could get a better insight into what is happening by looking at your data, you could use this to develop a follow-up experiment that helps to get to a better understanding which aspects of the differences drive the results.If for some reasons it is not possible for you to run more experiments, you should at least improve your analyses and discuss very thoroughly the limits of your design and thus, of your results.

We look forward to receiving your revised manuscript.

Kind regards,

Christiane Schwieren, Dr.

Academic Editor

PLOS ONE

Journal Requirements:

4. Please ensure that you refer to Figure 4 and 5 in your text as, if accepted, production will need this reference to link the reader to the figure.

Reviewers' comments:

Reviewer's Responses to Questions

**Comments to the Author**

1. Is the manuscript technically sound, and do the data support the conclusions?

Reviewer #1: Partly

Reviewer #2: Partly

2. Has the statistical analysis been performed appropriately and rigorously? 

Reviewer #1: I Don't Know

Reviewer #2: Yes

3. Have the authors made all data underlying the findings in their manuscript fully available?

Reviewer #1: No

Reviewer #2: Yes

4. Is the manuscript presented in an intelligible fashion and written in standard English?

Reviewer #1: Yes

Reviewer #2: Yes

5. Review Comments to the Author

Reviewer #1: This paper tries to extend the findings related to algorithm aversion to establish a connection between the consequences of a decision-making circumstance and the frequency of algorithm aversion. The authors conduct a lab experiment with 143 university students, which assume a businessperson's perspective who has to choose between an algorithm and a human expert to carry out a service. With six decision-making situations that vary in gravity, the authors found that the more severe consequences, the higher frequency participants exhibit algorithm aversion.

I sympathize with the efforts to continue understanding algorithm aversion. Still, I have concerns regarding the theoretical development for establishing the relationship (severity of the consequences – frequency of algorithm aversion) and the robustness of the findings due to limited experiments conducted. I will focus my comments on major and minor concerns in what follows.

Major concerns:

- The theory development relating the severity of the consequences of the decision and the presence of algorithm aversion needs more work. In the introduction, the authors briefly explain the relation mentioning the framing effect, but the relationship is not clear enough to the reader. In the derivation of hypothesis 3, the authors mentioned the model of bounded rationality, the cognitive limitations that humans suffer, and that great efforts are required when decisions have severe consequences. But why do decisions with more severe consequences induce less algorithm aversion, as Hypothesis 3 propose? Maybe the opposite could also be argued? Being the core of the paper, this relationship deserves a better explanation and discussion of the fundaments behind the possible behavior of decision-makers.

Maybe the authors can derivate their theory based on how risky people perceived the task. Some risk definitions may help, like the one proposed by Kaplan and Garrick (1981), considering the “triplet” definition of risk as “scenarios, probabilities and consequences.” Previous research has studied and suggests a higher presence of algorithm aversion in riskier circumstances (Dietvorst & Bharti, 2020; Grgic-Hlaca et al., 2019; Kawaguchi, 2020). For a general view, you can look at the high-level influencing factors of algorithm aversion described by Mahmud et al. (2022) in their literature review.

- A better explanation of why the six task were chosen would be useful related to the severity of the consequences. This may help considering as the authors mentioned, that in treatment B there is a larger range of severity evaluation, making some subjects assess the gravity of the decision as very high. Although authors ask how participants perceive the severity of the different tasks, finding significant differences between both treatment conditions, the tasks are in very different domains. Therefore, other phenomena could affect algorithm use rather than the severity of the consequences. Using control variables may help increase the robustness of the findings.

- Regarding the experiment conducted, I would prefer a bigger sample than 143 participants considering the experiment design. If we consider that in each treatment, there are three tasks and, if I understand correctly is a between design study, every task has around 24 participants. More explanation on sample sizing with power calculations would be helpful.

Minor concerns:

- The first paragraphs of the introduction focused on the definition of algorithm aversion. Although the discussion is valuable because I agree that different interpretations could be given to the concept “algorithm aversion” (Berger et al., 2020; Jussupow et al., 2020), I do not think it is the focus of this study. I suggest that the authors motivate in the introduction more directly to their main focus related to algorithm use and the connection between the consequences of a decision and the frequency of algorithm aversion.

- Maybe another title for the paper could guide readers better to what the paper is about rather than “The Tragedy of Algorithm Aversion.”

- Providing supplementary materials such as data, surveys, analyses conducted, and preregistrations (if available) would be helpful. This may help promote further research in related topics, as well as promoting open science practices. Maybe supplementary material is available, and I was not aware.

References

Berger, B., Adam, M., Rühr, A., & Benlian, A. (2020). Watch Me Improve—Algorithm Aversion and Demonstrating the Ability to Learn. Business and Information Systems Engineering, 1–14. https://doi.org/10.1007/s12599-020-00678-5

Dietvorst, B. J., & Bharti, S. (2020). People Reject Algorithms in Uncertain Decision Domains Because They Have Diminishing Sensitivity to Forecasting Error. Psychological Science, 31(10), 1302–1314. https://doi.org/10.1177/0956797620948841

Grgic-Hlaca, N., Engel, C., & Gummadi, K. P. (2019). Human Decision Making with Machine Assistance. Proceedings of the ACM on Human-Computer Interaction, 3(CSCW). https://doi.org/10.1145/3359280

Jussupow, E., Benbasat, I., & Heinzl, A. (2020). Why are we averse towards algorithms? A comprehensive literature review on algorithm aversion. ECIS 2020 Proceedings.

Kaplan, S., & Garrick, B. J. (1981). On The Quantitative Definition of Risk. Risk Analysis, 1(1), 11–27. https://doi.org/10.1111/J.1539-6924.1981.TB01350.X

Kawaguchi, K. (2020). When Will Workers Follow an Algorithm? A Field Experiment with a Retail Business. Management Science. https://doi.org/10.1287/mnsc.2020.3599

Mahmud, H., Islam, A. N., Ahmed, S. I., & Smolander, K. (2022). What influences algorithmic decision-making? A systematic literature review on algorithm aversion. Technological Forecasting and Social Change, 175. https://doi.org/https://doi.org/10.1016/j.techfore.2021.121390

Reviewer #2: The paper explores an interesting research question that has also been adequately derived from existing literature. Moreover, the study arrives at potentially meaningful results, should they be replicable. Unfortunately, there are some doubts about this due to problems with the experimental design.

Major comments:

Lack of control (I)

The experiment consists of 2 treatments, each treatment includes three “decision-making situations” (lines 119-130). This setup seems problematic as all six “situations” differ in many regards from each other (apart from the treatment effect “gravity of the consequences”). This relates to both differences between individual “situations” within one treatment as well as differences between individual “situations” across the two treatments and possible differences between treatments. Overall, these differences could provide alternative explanations for the observed effects that are not controlled for. In more detail:

• Treatment A (possibly serious consequences) consists of the three decision making situations “Autonomous driving”, “Evaluation of MRI scans” and “The assessment of criminal case files”, whereas treatment B (no serious consequences) consists of “Dating service”, “Selection of recipes” and “Drawing up weather forecasts”. The situational descriptions include many factors that could influence the decision of participants and that are often associated with algorithm aversion in the literature.

• For the description of the situations of treatment A this, for example, relates to factors such as morality and ethics (e.g. ethical questions related to decisions of autonomous vehicles in potential accident situations, morality questions in delegating criminal investigations). From the algorithm aversion literature it is also well understood that people in particular tend to distrust algorithms for medical diagnosis (here: MRI scan). In this situation, participants might in addition find it disturbing (or “immoral”) that the decision to rely on an algorithm is not made by a doctor but by the manager of the hospital (lines 511ff.). For the decision situations in treatment B these factors seem to be of less importance.

• Two of the situations of treatment B (dating services and receipts) appear to be more subjective in nature as compared to the situations of treatment A (in particular with regard to autonomous driving, but to a lesser extend also with regard to the other two “situations”). This also constitutes a difference to the less subjective third situation of treatment B (Weather forecast). This third “situation” of treatment B furthermore appears to differ in the degree of complexity. In addition, for weather forecasts participants might “expect” algorithms being involved. Overall, it seems that participants could potentially be more familiar with the use of algorithms in all three situations of treatment B. In the real world, algorithms are frequently used in such contexts already and real life experience of participants seems more probable here. Complexity, subjectivity and familiarity are all factors identified in the literature to affect algorithm aversion. This list is not exhaustive and more factors not controlled for might play a role.

Lack of control (II)

The authors explain that “The decision-making situations are selected in such a way that the subjects should be familiar with them from public debates or from their own experience. In this way, it is easier for the subjects to immerse themselves in the respective context.” (lines 134-137). Subjects will indeed very likely be familiar with the situation or context of the decisions from the real world (though to different degrees, as explained above). But this results in a loss of control over the experiment and therefore also appears problematic:

• The study defines a clear superiority of algorithms for all six “situations” identically (70% probability of success of the algorithm compared to 60% probability for human experts, lines 139-141). According to the definition of the authors, algorithm aversion only exists if subjects chose human experts despite the superior performance of algorithms. It appears problematic that better performance might not be given for the chosen situations, at least with regard to participants` experience and perceptions of the real world. For autonomous shuttle busses, for example, technological development is still at an early stage and the real world performance is often perceived as being (still) insufficient or at least poorer than that of a human operator. Choosing a human operator might thus be the result of this real-world experience or knowledge and not of algorithm aversion.

• With regard to the chosen (student) subject pool this problem seems to be of particular relevance. Some of the decision situations seem to be directly linked to the content of the study programs of participants (autonomous driving/ health diagnosis). 60 subjects (42.0%) study at the Faculty of Vehicle Technology, and 18 subjects (12.6%) at the Faculty of Health Care (lines 201 and 202).

Problems related to the incentive structure

• With the chosen experimental design, it remains somewhat unclear whether subjects base their decision on the situational description or on the performance factor/ the success probabilities provided in the introduction of the experiment.

• Participants could either understand the missing link between incentive scheme and situational descriptions. In this case it seems probable that some participants understand the game as a choice between two lotteries and decide solely based on the probabilities of the lotteries ignoring the context completely. Or they could base their decisions on the situational descriptions. In this case it would seem likely that participants are influenced by real world experience and not only (if at all) by the success probabilities of the card decks provided. The perceived real world performance might contradict with the probabilities provided. The heterogeneous background of the participants (different study programs that are related to the scenarios) might also play a role here.

• The lotteries are implemented with the help of physical card decks (lines 214/215). This non-digital implementation of the lotteries might further support the “decoupling” of incentive and situational description in this particular context (algorithms).

• Probabilities are “made up” and not created within the experiment.

• It should be noted that the success probabilities are explained in the introduction of the experiment, but not in the situational descriptions. This might affect the salience of this information.

• The incentive scheme does not take differences in gravity into account.

• Lines 336-337: “The differing consequences of the decision-making situations do not affect the subjects themselves, but possibly have implications for third parties.” Such implications for third parties are also purely hypothetical and outside the incentive scheme of the experiment.

Data analysis

The description and analysis of the data seems somewhat incomplete:

• Line 203: How many participants for the individual scenarios?

• Lines 226ff. Gravity check: no information provided on how participants assessed the gravity of the six individual scenarios. The scenarios within a treatment are added up without any further analysis. Differences between the scenarios could also (partially) explain why “subjects perceive the gravity of the decision-making situations significantly differently”.

• Analyze for possible differences resulting from different backgrounds (study programs)

• Analyze for possible gender effects (minor comment)

• Discuss the results of the manipulation check in more detail. Did manipulation really work properly for all six situations (e.g. recipes)?

To conclude: the authors argue that the “decisive advantage of a framing approach is that the influence of a factor can be clearly identified. There is only one difference between the decision-making situations in Treatment A and Treatment B: the gravity of the possible consequences.” (lines 306 -308). I cannot fully agree with this statement. All six situations differ from each other by more than one factor. Also, all three situations of treatment A taken together seem to differ by more than one factor from the situations of treatment B taken together. Some of these factors have been identified in previous literature as being particularly linked to algorithm aversion. In addition, the incentive scheme is not fully convincing. As a result, it to some degree remains unclear whether the observed differences are the result of the treatment effect or of other (uncontrolled for) differences or of real world experience. Some of these problems could possibly be addressed by a more in depth analysis of the data (the regression analysis described in lines 277ff. indicates a promising direction as it does not rely on the aggregation of scenarios into treatments), whereas some of the limitations seem inherent to the experimental design and should, at least, be discussed as such.

Minor comments

• Line 41: for a very recent systematic literature review also see: Hasan Mahmud, A.K.M. Najmul Islam, Syed Ishtiaque Ahmed, Kari Smolander, What influences algorithmic decision-making? A systematic literature review on algorithm aversion, Technological Forecasting and Social Change, Volume 175, 2022, 121390.

• Line 64/table 1: Sometimes a definition is provided that is only an indirect quote of a definition already provided elsewhere in the table. The added value of doing so remains somewhat unclear.

• Lines 93-94: More careful formulation suggested with regard to normative recommendations in particular when other aspects (e.g. ethical considerations) may also be of importance.

• Test questions: Explain what happens if someone answers test questions incorrectly (I assume that question needs to be answered again, but this is not explicitly mentioned in the manuscript).

6. PLOS authors have the option to publish the peer review history of their article (what does this mean?). If published, this will include your full peer review and any attached files.

Reviewer #1: No

Reviewer #2: No

---

## [Author Response · Author response to Decision Letter 0]

26 Aug 2022

The reviewers have sent us numerous suggestions for improvement. Unfortunately, our response letter is thus too long to fit in this text box in the submission tool. We have therefore uploaded it as a separate MS Word document and ask you to take a look at our responses there.

---

## [Decision Letter · Decision Letter 1]

26 Sep 2022

PONE-D-21-39459R1The Extent of Algorithm Aversion in Decision-making Situations with Varying GravityPLOS ONE

Dear Dr. Lorenz,

Thank you for submitting your manuscript to PLOS ONE. After careful consideration, we feel that it has merit but does not fully meet PLOS ONE’s publication criteria as it currently stands. Therefore, we invite you to submit a revised version of the manuscript that addresses the points raised during the review process.

 From my own reading, both reviewers have made very careful and partially also similar comments, and found them partially addressed.Even though I see that adding more data is impossible at this stage (it would have been a major improvement), being more clear in the description of the theory behind the hypotheses is, in my view, key. Adding literature is one thing, that is certainly an improvement, but considering this literature thoroughly would improve the paper even more.The same holds for the discussion of design choices that might not be considered ideal in hindsight. Every experiment has design choices that, after the fact, are considered not ideal. However, if there are ways to deal with that in the data analysis - if only in being very careful with respect to causal language - is warranted, if it is impossible to add to the data collection.I also see some contradictions in the argumentation of the authors with respect to the vignettes. It is clearly NOT the case that the only thing that changes is gravity - it is always context AND gravity that changes, and thus, the results are less clear than the authors claim.I can imagine that the authors might consider this too much change to the paper, but even though Plos One differs in publication criteria from other outlets, it does have a strong commitment to clear and correct data analysis and interpretation of the results - including to be careful not to oversell.

We look forward to receiving your revised manuscript.

Kind regards,

Christiane Schwieren, Dr.

Academic Editor

PLOS ONE

Reviewers' comments:

Reviewer's Responses to Questions

**Comments to the Author**

1. If the authors have adequately addressed your comments raised in a previous round of review and you feel that this manuscript is now acceptable for publication, you may indicate that here to bypass the “Comments to the Author” section, enter your conflict of interest statement in the “Confidential to Editor” section, and submit your "Accept" recommendation.

Reviewer #1: (No Response)

Reviewer #2: (No Response)

2. Is the manuscript technically sound, and do the data support the conclusions?

Reviewer #1: Partly

Reviewer #2: Partly

3. Has the statistical analysis been performed appropriately and rigorously? 

Reviewer #1: I Don't Know

Reviewer #2: Yes

4. Have the authors made all data underlying the findings in their manuscript fully available?

Reviewer #1: Yes

Reviewer #2: Yes

5. Is the manuscript presented in an intelligible fashion and written in standard English?

Reviewer #1: Yes

Reviewer #2: Yes

6. Review Comments to the Author

Reviewer #1: I want to thank the authors for their work, but I do not think the current status of the paper meets the publication standard required. Reading the responses to my comments and the revised manuscript, I realize that the authors have not adequately addressed my major concerns. Particularly my major concerns regarding the theory development is not well explained in the introduction, and only adding references to the derivation of hypothesis 3 is not enough to clarify the theory relationship between severity of the consequences and algorithm aversion. Also, I think that 143 participants is a very small sample, as I mentioned before, and probably the experiment is underpowered. Unfortunately, the authors were not able to conduct new experiments. Therefore, I think this paper requires significant rework and especially conducting more experiments to get published.

Reviewer #2: The authors have considered some of the comments and have improved the paper considerably. However, this does not fully apply to the main problem I have seen (lack of control/ risk of spurious results). In contrast to the authors' explanation, I am still not fully convinced that the design allows for a clear proof of causality with respect to the correlation between gravity and algorithm aversion.

A causal relationship between two variables exists if a variation in the independent variable results in a variation in the dependent variable, keeping all other things equal (ceteris paribus). As explained in my previous comments, the scenarios (or treatments) are quite different from each other, they differ in many aspects. The authors have included a sentence addressing this difference now. But they, in my view, still do not recognize that as a result of these differences the relationship between the two variables could be spurious/ due to changes in a third variable.

In the reply letter, the authors (if understood correctly) imply that their experiment constitutes a vignette study (author response to Reviewer 2 comment 9). I also do not find this argument convincing. While I agree with the authors that a vignette design could proof useful to answer the RQ, I do not believe that such a design has been implemented (the term “vignette” does also not appear in the paper). Vignette surveys (factorial surveys) include a description of a situation, consisting of a systematic combination of characteristics (“dimensions”) and a systematic variation of these dimensions (“levels”). Through experimental variation unconfounded effects of the dimensions/ factors can then be estimated. The experiment on hand does, however, not vary the dimension “gravity” within one scenario (e.g. MRT scans for live threatening diagnosis vs. MRT scans for less severe diagnosis; weather forecast with possible live threatening consequences, e.g. for sailors vs. weather forecast with less severe consequences such as a beach day which is cancelled), but compares across different scenarios. This difference to the standard vignette experimental design is (again) problematic with regard to causality.

As explained earlier, this problem seems inherent to the experimental design to me. I believe that the experiment can still add value to the literature, but that this limitation (if correctly observed and shared by the other reviewer) is still not addressed sufficiently in the paper.

7. PLOS authors have the option to publish the peer review history of their article (what does this mean?). If published, this will include your full peer review and any attached files.

Reviewer #1: No

Reviewer #2: No

---

## [Author Response · Author response to Decision Letter 1]

6 Nov 2022

Please refer to the MS Word document that we have uploaded with our revised manuscript.

---

## [Decision Letter · Decision Letter 2]

23 Nov 2022

The Extent of Algorithm Aversion in Decision-making Situations with Varying Gravity

PONE-D-21-39459R2

Dear Dr. Lorenz,

We’re pleased to inform you that your manuscript has been judged scientifically suitable for publication and will be formally accepted for publication once it meets all outstanding technical requirements.

Kind regards,

Christiane Schwieren, Dr.

Academic Editor

PLOS ONE

Additional Editor Comments (optional):

Reviewers' comments:

Reviewer's Responses to Questions

**Comments to the Author**

1. If the authors have adequately addressed your comments raised in a previous round of review and you feel that this manuscript is now acceptable for publication, you may indicate that here to bypass the “Comments to the Author” section, enter your conflict of interest statement in the “Confidential to Editor” section, and submit your "Accept" recommendation.

Reviewer #1: (No Response)

Reviewer #2: All comments have been addressed

2. Is the manuscript technically sound, and do the data support the conclusions?

Reviewer #1: Partly

Reviewer #2: (No Response)

3. Has the statistical analysis been performed appropriately and rigorously? 

Reviewer #1: I Don't Know

Reviewer #2: (No Response)

4. Have the authors made all data underlying the findings in their manuscript fully available?

Reviewer #1: Yes

Reviewer #2: (No Response)

5. Is the manuscript presented in an intelligible fashion and written in standard English?

Reviewer #1: Yes

Reviewer #2: (No Response)

6. Review Comments to the Author

Reviewer #1: I appreciate the authors' efforts regarding the derivation of the hypotheses and the further analysis using cluster analysis. Still, as I mentioned previously, it is necessary to conduct more experiments mainly to increase the number of participants. Therefore, if no further experiments are possible, I think this paper does not meet the threshold to be accepted.

Reviewer #2: (No Response)

7. PLOS authors have the option to publish the peer review history of their article (what does this mean?). If published, this will include your full peer review and any attached files.

Reviewer #1: No

Reviewer #2: No

---

## [Editor Report · Acceptance letter]

28 Nov 2022

PONE-D-21-39459R2 

The Extent of Algorithm Aversion in Decision-making Situations with Varying Gravity 

Dear Dr. Lorenz:

I'm pleased to inform you that your manuscript has been deemed suitable for publication in PLOS ONE. Congratulations! Your manuscript is now with our production department. 

Kind regards, 

on behalf of

Dr. Christiane Schwieren 

Academic Editor

PLOS ONE